# Feasibility Analysis and Implementation of Head-Mounted Electrical Impedance Respiratory Monitoring

**DOI:** 10.3390/bios12110934

**Published:** 2022-10-27

**Authors:** Hongli Yan, Xudong Yang, Yanyan Liu, Wanting He, Yipeng Liao, Jiejie Yang, Yueming Gao

**Affiliations:** 1College of Physics and Information Engineering, Fuzhou University, Fuzhou 350108, China; 2The International Joint Laboratory on Intelligent Health Monitoring Systems, Fuzhou University, Fuzhou 350108, China; 3Key Lab of Medical Instrumentation & Pharmaceutical Technology of Fujian Province, Fuzhou 350108, China; 4The School of Advanced Manufacturing, Fuzhou University, Quanzhou 362251, China

**Keywords:** respiratory monitoring, wearable devices, bioelectrical impedance, health monitoring

## Abstract

The respiratory rate is one of the crucial indicators for monitoring human physiological health. The purpose of this paper was to introduce a head-mounted respiratory monitoring solution based on electrical impedance sensing. Firstly, we constructed a finite element model to analyze the feasibility of using head impedance for respiratory sensing based on the physiological changes in the pharynx. After that, we developed a circuit module that could be integrated into a head-mounted respiratory monitoring device using a bioelectrical impedance sensor. Furthermore, we combined adaptive filtering and respiratory tracking algorithms to develop an app for a mobile phone. Finally, we conducted controlled experiments to verify the effectiveness of this electrical impedance sensing system for extracting respiratory rate. We found that the respiration rates measured by the head-mounted electrical impedance respiratory monitoring system were not significantly different from those of commercial respiratory monitoring devices by a paired *t*-test (*p* > 0.05). The results showed that the respiratory rates of all subjects were within the 95% confidence interval. Therefore, the head-mounted respiratory monitoring scheme proposed in this paper was able to accurately measure respiratory rate, indicating the feasibility of this solution. In addition, this respiratory monitoring scheme helps to achieve real-time continuous respiratory monitoring, which can provide new insights for personalized health monitoring.

## 1. Introduction

All human organs require oxygen from breathing to produce energy, indicating that breathing is essential to maintain the proper functioning of the physiological systems in our bodies. Respiratory monitoring helps to understand the health status of humans, which can be used for disease prevention and diagnosis [1,2,3]. Studies have shown that respiratory information can help diagnose respiratory diseases [4] and diseases of the nervous system, cardiovascular system [5], and excretory system [6]. Meanwhile, respiratory information can also be applied to health and safety monitoring areas such as driving [7] and the military. In addition, respiratory information is available for emotion analysis [8,9] and entertainment interaction [10]. Respiratory rate is one of the most critical indicators of respiratory information. In general, adults’ breaths per minute (BPM) are 12 to 20 [11], whereas the BPM of the elderly are 10 to 30 [12]. Achieving real-time, dynamic, and continuous monitoring of respiratory rate parameters is of great importance for personalized medicine and a higher level of healthcare.

Respiration relies mainly on chest expansion [13], accompanied by gas exchange, which affects the surrounding air’s temperature [14], humidity [15], and composition. Most of the existing respiratory monitoring methods depend on sensors to acquire the undulating changes in chest morphology or gas exchange. The monitoring devices are classified as contact and noncontact according to how the sensors are deployed. Contact respiratory monitoring has been widely used for devices in clinical settings [16], such as spirometry, capnography, and impedance spirometry. However, specialized respiratory monitors suffer from being bulky and less portable. The wearable respiratory monitoring devices that have emerged in recent years have made significant progress in optimizing wearing comfort [17]. In the context of the coronavirus disease 2019 epidemic, masks have become indispensable in daily life for epidemic prevention. Mitar Simić et al. [18]. proposed respiratory monitoring masks with embroidered capacitive sensors. Fan, W. et al. [19]. designed respiratory monitoring textiles with woven washable sensor arrays to prolong the use of sensors. However, as frequently changed and washed items, the textiles need to be equipped with multiple sets for daily use, which may increase the burden on patients and cause cross-contamination. Noncontact respiratory monitoring is mainly done by deploying microphones [20], radar [21], or camera sensors [22] in space to capture information such as human breath sounds [23], chest changes [7,24], and facial thermography. Although these methods enable noncontact monitoring, they face new challenges regarding privacy protection, multitarget monitoring, noise filtering, and measurement accuracy.

Respiration leads to changes in gas volume and associated muscle morphology in the body, resulting in subsequent changes in the electrical impedance of muscles in multiple parts of the respiratory system. Therefore, bioelectrical-impedance-sensing-based measurement is a promising technique for respiratory monitoring. Kaan Sel et al. [25] developed a wearable clothing respiratory monitoring system by integrating multiple sensors and improved the system measurement accuracy by processing multisensor data with the help of data fusion algorithms. Chunkai Qiu et al. [26] designed an IoT-connected wearable chest patch based on electrical impedance changes in the chest to achieve real-time continuous respiratory monitoring in daily life, significantly improving users’ comfort. Emanuele Tavanti et al. [27] reported a new method of respiratory monitoring based on electrical impedance measurements, proposing to use physiological changes in the pharynx to measure respiratory rate. The scheme provided a new idea for wearable electrical impedance respiratory monitoring. However, collecting muscle impedance data by surface electrodes is prone to electrode dislodgement. Moreover, previous signal processing algorithms have relied on computer programs such as MATLAB for offline analysis, which means that respiratory rate cannot be provided in real time. Therefore, more researchers must work towards realizing real-time continuous respiratory monitoring and improving wearable respiratory monitoring devices.

A head-mounted respiratory monitoring scheme is presented in this paper. The scheme uses electrical impedance sensors to perceive impedance changes due to physiological changes in the pharynx during respiration, allowing respiratory frequency extraction. The scheme embeds electrodes in a head-mounted device placed under the mastoid bone on both sides of the head, measuring the change in head (pharynx) impedance caused by respiration to perceive the user’s respiratory pattern and frequency. As shown in Figure 1, this respiration monitoring system can be integrated into helmets, glasses, and other wearable devices in daily life with the advantages of comfortable wearing and excellent experience, making it widely used in scenarios such as transportation, entertainment, and daily life.

The remaining of this paper is organized as follows: Section 2 analyzes the feasibility of the pharyngeal impedance respiratory monitoring method based on the anatomical structure and experiments. Section 3 describes the implementation process of the head-mounted electrical impedance respiratory monitoring system in detail. Section 4 validates the pharyngeal impedance respiratory monitoring method and system performance proposed in this paper through controlled experiments. Section 5 compares the existing wearable devices, discusses the advantages and limitations of the proposed system, and looks forward to future research ideas. Conclusions are given in Section 6.

## 2. Feasibility Analysis

### 2.1. Anatomical Structure Analysis

The respiratory system consists of two parts, the respiratory tract and the lungs, where the respiratory tract is the pathway for gas transport. During gas transport, the increase of the gas volume in the respiratory tract and the widening of the electrical conductivity path lead to an increase in electrical impedance. Therefore, respiratory activity can be perceived via bioelectrical impedance. As shown in Figure 1, the respiratory tract located in the head includes the nasal cavity and the pharynx. Various bones encase the nasal cavity, and the change in electrical resistance within the nasal cavity during respiratory activity is fragile due to the relatively poor electrical conductivity of the bones. As shown in Figure 2a, the pharynx is located below the mastoid bone and is wrapped by muscles. Due to the muscle’s better electrical conductivity, it can conduct the electrical impedance changes in the pharynx during respiratory activity better. In particular, the relatively large amplitude of periodic expansion and contraction of the Eustachian tube leads to a periodic change in the electrical conductivity distribution within the head over time. As shown in Figure 2b, the surface electrodes can be placed near the mastoid bone of the head. Therefore, bioelectrical impedance changes in the subpapillary pharynx of the mastoid bone measured by surface electrodes can be used to capture respiratory activity. At the same time, the mastoid position enables an easy integration of the measurement electrodes into wearable devices such as helmets and glasses, providing a better user experience for achieving continuous respiratory monitoring.

### 2.2. Simulation Analysis

To evaluate the feasibility of monitoring respiratory rate with the measurement of pharyngeal impedance changes, a simple multilayer head geometry model was constructed in the AC/DC module of COMSOL Multiphysics 5.4 based on the anatomical structure of the head in this paper. As shown in Figure 2c,d, the model consisted of skin, skull, brain, muscle, pharynx, trachea, and surface electrodes.

The head was presented as an overall ellipsoidal structure (a = 16 cm, b = 13 cm, c = 13 cm), where the skin thickness was 2 mm, the skull thickness was 1 cm, and the filling within the skull was assumed to be a homogeneous medium. The dielectric properties of each tissue, namely relative permittivity and conductivity, were obtained from published literature and online reference databases [28,29,30], and their specific parameters were set as shown in Table 1.

For simulating the periodic changes of Eustachian volume over time during respiration, the volume was calculated from the paper of Emanuele Tavanti [7] with the following equation:(1)$$d_{p}\left(t\right)=\frac{d_{p,max}-d_{p,min}}{2}\left[1-\cos\left(2\pi f_{r}t\right)\right]+d_{p,min}$$

The pharynx is assumed to be an ellipsoid $d_{p}=\left\{a,b,c\right\}$, *a* and *b* are the equatorial radius, *c* is the polar radius, and $f_{r}$ is the respiratory frequency.

In the simulation model, we assumed that the equatorial radius of the pharynx a = 7.3∼8.7 mm, b = 13.8∼16.3 mm, the pole radius c = 25 mm, and the respiratory frequency $f_{r}$ = 20 bpm. According to the study by Schwartz, S. [31], muscles are often most responsive at 50 kHz. We modified the volume of the pharynx according to Equation (Equation 1), which was used to simulate six cycles of respiratory activity. Simultaneously, an excitation current of 1 mA and 50 kHz was added to the electrode below the mastoid bone. We measured the voltage between the two electrodes and calculated the impedance, the results of which were shown in Figure 3. Due to the geometric changes in the pharynx caused by respiration, we found that the head impedance exhibited a sinusoidal waveform similar to that of respiration. Figure 2d shows the cross-sectional current density mode distribution of the head. The value of the current density mode in the pharynx was zero, which indicated that the pharynx could indeed impede the current and thus affect the head impedance. This result was consistent with the results analyzed in Section 2.1. Changes in the morphology of the Eustachian muscles during respiration cause changes in bioelectrical impedance, indicating that appropriately designed impedance acquisition devices can be used for monitoring respiratory activity.

## 3. Systems Design

To validate the simulation results and implement head-mounted electrical impedance respiration monitoring, we independently designed an electrical impedance sensing system for respiration monitoring, which could be integrated into helmets, VR glasses, and eyeglasses. As shown in Figure 4, the system mainly included electrical impedance sensing and signal processing modules. Two surface electrodes were attached to both sides of the head under the mastoid bone to measure the change in head impedance and sense the pattern and frequency of each breath. The choice of this site for impedance measurements had two advantages. First, monitoring the impedance changes caused by respiratory activity was easier. The second was that the measurement electrodes at this site were easy to integrate into a head-mounted wearable device, which could improve the consumer’s experience. The impedance data measured by the electrical impedance sensing module were transmitted wirelessly via Bluetooth to an intelligent terminal app after a microprocessor’s impedance calculation and preprocessing the signal. We developed adaptive filtering for noise reduction and respiratory tracking algorithms on the app for real-time calculation and visualization of the respiratory frequency.

### 3.1. Electrical Impedance Sensing

The electrical impedance sensing module employed the AD5933 high-precision impedance converter of Analog Devices for detecting respiration-induced changes in pharyngeal impedance. The DDS output signal was implemented via $I^{2}C$ programming to excite the external complex impedance. The response signal of the head impedance was sampled by the on-chip ADC and then processed by the on-chip DSP using a discrete Fourier transform (DFT).
(2)$$X\left(f\right)=\sum_{n=0}^{1023}\left(x\left(n\right)\left(\cos(n)-j\sin(n)\right)\right)$$
where X(f) indicates the energy of the signal at frequency *f*, x(n) denotes the output of the ADC, cos(n) and sin(n) are the sample test vectors at frequency *f* provided by the DDS core.

Real and imaginary data were returned for each frequency after the DFT processing of the sampled signal. After $I^{2}C$ reading and calibration calculation, the impedance amplitude and relative phase at the scan frequency point could be calculated. The results were transmitted to an intelligent terminal app via low-power Bluetooth. The AD5933 supported four voltage excitation mode outputs, each with a different direct current (DC) bias and output impedance. When measuring small impedance signals, the high DC output impedance of VOUT at the output and the transimpedance amplifier of VIN at the receiver may generate saturation distortion, resulting in excessive current flow through the body and creating a safety hazard. Therefore, we designed an additional analog front-end circuit by combining data from technical manuals and the literature [26]. Its measurement circuit is shown in Figure 5. Capacitors $C_{1}$ and $C_{2}$ were added at the output to isolate the DC bias and adjust the DC bias to VCC/2. By adjusting the ratio of resistors $R_{1}$ and $R_{2}$ as well as the value of voltage follower $R_{out}$, the maximum output current of the electrical impedance sensing module was made to meet the human body safety guidelines and achieve higher precision for small impedance measurements.

To achieve high-accuracy impedance measurements, it is critical to determine a reasonable AD5933 gain factor(GF). When calculating the GF, the receiver stage of the AD5933 must work within its linear interval.
(3)$$V_{ADC\_MAX}⩾V_{OUTPUT}\times \frac{Z{}_{RFB}}{Z_{UNKNOWN}}\times PGA$$
where $V_{ADC\_MAX}$ is the analog reference voltage of the internal ADC in the AD5933, $V_{OUTPUT}$ is the output excitation voltage, $Z_{UNKNOWN}$ is the impedance to be measured, and PGA is the amplification of the input response signal in the ADC.

According to Equation (Equation 3), we designed the analog front-end circuit in combination with the range of impedance values of the head. It was used for consideration of the gain of the overall system to meet the linear operating area of the ADC sampling. Based on the head impedance values derived from the finite element simulation in Section 2.2 and the available research literature, we determined a head impedance value of approximately 560 Ω for two-electrode measurements. Therefore, we chose a calibration impedance value of 560 Ω when calculating the gain factor of the AD5933. It was calculated as follows:(4)$$GF=\frac{Admittance}{Code}=\frac{\frac{1}{Z_{CALIBRATION}}}{M}$$
where *M* is the magnitude value at the corresponding frequency point after the DFT transform of AD5933. $Z_{CALIBRATION}$ is the calibration impedance. The impedance calculation formula is as follows:(5)$$\left|Z\right|=\frac{1}{GF\times M}$$

### 3.2. Signal Processing

As described in Section 2, the periodic movement of the pharynx during respiratory activity generates a trend in head impedance over time similar to that of respiration. Thus, the respiratory frequency can be obtained by extracting the signal associated with the respiratory activity in head impedance measurements. However, the periodic signals associated with respiration were masked in various noises during the measurement, including environmental noise, blood circulation system, and motion disturbances. Due to the random characteristics of the noise signals, it may not be easy to extract the respiration rate accurately due to the overlapping phenomenon between the respiration signals and the noise signals. To this end, we designed and implemented an adaptive filtering tracking algorithm to extract the desired respiratory signal from the time series signals of head impedance measurements.

#### 3.2.1. Adaptive Filtering for Noise Reduction

Firstly, we normalized the measured head impedance data in terms of magnitude to obtain the time series x(n). Then, we designed a filter of order *N* with parameters W(n), and the output was as follows:(6)$$y\left(n\right)-\sum_{i=0}^{N-1}w_{i}\left(n\right)x\left(n-1\right)=W^{{}^{T}}\left(n\right)X\left(n\right)=X^{{}^{T}}\left(n\right)W\left(n\right)$$
where X(n) and W(n) were calculated by the following equations:(7)$$X\left(n\right)={\left[x\left(n\right),x\left(n-1\right),\ldots x\left(n-N+1\right)\right]}^{T}$$
(8)$$W\left(n\right)={\left[w_{0}\left(n\right),w_{1}\left(n\right),\ldots w_{N-1}\left(n\right)\right]}^{T}$$

Assuming that the desired output signal was d(n), the formula for calculating the error signal was as follows:(9)$$e\left(n\right)=d\left(n\right)-y\left(n\right)=d\left(n\right)-W^{{}^{T}}\left(n\right)X\left(n\right)$$

According to the minimum mean square error criterion, the minimization objective function J(W) was calculated to be as follows:(10)$$J\left(W\right)=E\left\{{\left|e{\left(n\right)}^{}\right|}^{2}\right\}=E\left\{{\left|d\left(n\right)-W^{T}\left(n\right)X\left(n\right)\right|}^{2}\right\}$$

For computational convenience, we replaced the mathematical expectation of *J*(*W*) with the instantaneous gradient.
(11)▽(n)=−2e(n)X(n)

Finally, we could obtain the iterative update formula in the standard time domain as follows:(12)W(n+1)=W(n)+2μX(n)e(n)

The filter coefficients were updated whenever a new x(n) and d(n) were given, where μ denoted the update step. The best-estimated output y(n) was obtained by an iterative error calculation.
(13)$$Y\left(n\right)=\left[y_{1},y_{2},\ldots y_{N}\right]$$

#### 3.2.2. Calculation of Respiration Rate

The respiration rate extraction window interval was set as $\left[N_{i},N_{i}+2*T_{min}*F_{s}\right]$ to ensure that at least two breaths could be detected within this interval, where $N_{i}$ represents the starting position and $F_{s}$ denotes the sampling rate. Afterward, we found all the wave peaks in the interval and recorded their indexes.
(14)$$If\left\{\begin{array}{c} y(k)>y(k+1)\\ y(k)>y(k-1) \end{array}\right.,k\in \left\{N_{i}+1,N_{i}+2,\ldots N_{i}+2\cdot T_{min}\cdot F_{s}\right.\Rightarrow P_{max}^{n}=\left\{\begin{array}{c} val=y(k)\\ idx=k \end{array}\right.$$
where $P_{max}^{n}$ is the storage array of the wave peaks, val indicates the amplitude of the wave peaks, and idx denotes the index value of the wave peaks in the y(n) sequence.

In addition, we also found all the wave troughs in the interval and recorded their indexes as follows.
(15)$$If\left\{\begin{array}{c} y(k)<y(k+1)\\ y(k)<y(k-1) \end{array}\right.,k\in \left\{N_{i}+1,N_{i}+2,\ldots N_{i}+2\cdot T_{min}\cdot F_{s}\right.\Rightarrow P_{min}^{n}=\left\{\begin{array}{c} val=y(k)\\ idx=k \end{array}\right.$$
where $P_{min}^{n}$ is the storage array of the wave troughs, val represents the amplitude of the wave troughs, and idx indicates the index value of the wave troughs in the y(n) sequence.

Finally, we calculated the mean value of the index difference between adjacent wave peaks and wave troughs.
(16)$$If\left\{\begin{array}{c} P_{max}^{k}.val>0\\ P_{min}^{k}.val<0 \end{array}\right.,k\in \left\{N_{i},N_{i}+1,\ldots N_{i}+2\cdot T_{min}\cdot F_{s}\right.\Rightarrow \Delta idx=\frac{\left(P_{max}^{k}-P_{max}^{k-1}\right)+\left(P_{min}^{k}-P_{min}^{k-1}\right)}{2}$$

Therefore, the respiration rate can be calculated by the formula as follows:(17)$$RR=60\times \frac{F_{s}}{\Delta idx}$$

#### 3.2.3. Electronic Devices and Software Processes

As shown in Figure 5a, we employed STM32F103RCT6 from STMicroelectronics as the microprocessor unit. The Bluetooth 4.0 module HC-05 from Guangzhou Huicheng Information Technology Co., Ltd. (Guangzhou, China) was selected for the Bluetooth module. The hardware PCB design and fabrication used EasyEDA from Shenzhen JLC Electronics Co., Ltd. (Shenzhen, China) (https://lceda.cn/, accessed on 23 October 2022). The battery adopted the 5V1A Boosting Lithium Batteries (1000 mAh) produced by ZONCELL INTERNATIONAL LIMITED (Shenzhen, China). The circuit device was encapsulated in a protective case and had an overall size of 68 × 43 × 18 mm, achieving a light weight. Figure 5b presents the flow chart of the embedded piece of software for the electronic device. The core process was implemented based on the principles introduced in Section 3.1 and developed in the C language. In addition, the app was developed in the java language on Android Studio 4.0.1 platform based on the principles introduced in Figure 4 and Section 3.2.

## 4. Experiments and Results

### 4.1. Experimental Protocol

This experimental protocol was filed and approved by the Key Laboratory of Medical Devices and Pharmaceutical Technology in Fujian Province. The impedance data from 16 healthy young subjects in the respiratory experiment were collected in a stable indoor environment. The results were used to initially evaluate the feasibility of the head-mounted electrical impedance respiratory monitoring system proposed in this paper. Firstly, the recruited subjects were briefed in detail about the practical steps and signed an informed consent form for the experiment. After a subject was seated at the assigned position, the skin below the mastoid bone on both sides of the subject’s head was wiped with alcohol swabs, and crescent-shaped gel Ag/AgCl physiotherapy electrodes (Jing dian yi kang, LD-1) were attached to the area, which was shown in Figure 2b. Then, we connected the circuit module of the electrical impedance respiratory sensing system to the physiotherapy electrodes. Furthermore, we started the mobile app to complete the Bluetooth connection and set the excitation frequency of the measurement to 50 kHz and the sampling frequency to 80 Hz. Meanwhile, the subject was asked to wear a commercial airflow sensing-based sleep breathing monitor (SNORE CIRCLE, Y20). Finally, we started the experiment by measuring the data when the subjects maintained a steady breathing state, and the acquisition time was about 3 min for each subject. After the experiments, the collected head impedance data were processed by an adaptive filtering algorithm for noise reduction, and the respiratory frequency was calculated accordingly. We used the respiratory rate recorded by Y20 as a reference value and compared it with the calculated respiratory frequency for analysis.

### 4.2. Experimental Results

#### 4.2.1. Noise Reduction Processing

Figure 6 demonstrates the head impedance signal acquired by subject one at the one-minute duration in the respiration experiment. We found a lot of high-frequency noise disturbance in the collected impedance data from Figure 6a. Moreover, the waveform of the head impedance signal caused by respiration was almost drowned, and only the general trend could be seen. Therefore, extracting the respiration frequency directly from the original impedance data was difficult. As shown in Figure 6b, we used a time–frequency analysis toolbox of Matlab to analyze the original signal and plot the wavelet scale. We found a spectrum of signals near the frequency at 0.3 Hz for the selected signal, consistent with the finite element simulation results. This frequency corresponded to the respiratory frequency of adults, indicating that the head impedance data could indeed characterize the respiratory signals. Therefore, we had to filter out the noise disturbances at other frequencies and then extract a precise impedance characterization for the respiratory waveform.

It is known that the standard respiratory frequency of adults is 12–20 bpm, and the respiratory waveform is close to a sinusoidal waveform. Figure 7 illustrates the impedance signal after noise reduction processing in the respiration experiment. We employed adaptive filtering to denoise the original data and normalized the amplitude. The frequency of the reference sine wave for the adaptive filter was set to 0.3 Hz, and the filter step was 0.001. As shown in Figure 7a, a complete waveform of the respiration signal could be restored after adaptive filtering the raw data. Figure 7b also presents the time–frequency analysis results of the impedance data in the respiration experiment after adaptive filtering. We noticed that only one straight line coinciding with the respiration frequency remained in the wavelet scale plot, indicating that the adaptive filter used in this paper worked effectively.

#### 4.2.2. Performance Evaluation

To evaluate the reliability of the head-mounted electrical impedance respiratory monitoring system, we performed adaptive filtering on the head impedance data of subject one and extracted the respiratory waveform. As shown in Figure 8, we compared the extracted respiratory waveform from the impedance data with the reference respiratory waveform from the Y20 device. We found that the head-mounted electrical impedance respiratory monitoring device could accurately detect the respiratory waveform from the subject. Although there was a slight difference in amplitude between the respiratory waveform from the impedance data and the respiratory waveform measured by the Y20, the correlation coefficient between them was as high as 0.9965. It indicated that the respiratory waveform could be extracted from the head impedance data, and the respiratory frequency could also be calculated.

Table 2 presents statistical information on the simultaneous respiratory monitoring of 16 subjects by both devices. The mean respiratory rates measured with the head-mounted electrical impedance respiratory monitoring device and the Y20 were 18.871 and 18.584, respectively. After using the paired *t*-test, it was shown that there was no significant difference between the respiration rates measured by the two devices (*p* = 0.111 > 0.05). Moreover, the 95% confidence interval of the difference between the two devices was −0.680 to 1.055. Additionally, there was no significant deviation from zero for the difference between the two devices, providing excellent consistency.

Figure 9 exhibits the Bland–Altman distribution of respiration rates measured by the two devices for 16 subjects. We noticed that the overall difference in respiration rates measured by both devices was distributed around zero, and all respiration rates were within the 95% interval. It indicated an excellent consistency between the respiration rates measured by the two devices. Therefore, the method of electrical impedance respiration monitoring proposed in this paper had a good reliability, which was suitable for head-mounted respiration monitoring scenarios and provided a good experience for users.

## 5. Discussion

Existing wearable respiratory monitoring devices have made significant progress in being comfortable to wear and coping with complex monitoring scenarios while ensuring high-accuracy respiratory rate measurements. Table 3 summarizes and compares some of the latest wearable respiratory monitoring systems. The current research points focus on the improvement of new materials and sensors. Respiratory monitoring sites include the mouth, nose, chest, and abdomen, with little exploration of respiratory monitoring in the pharynx. There is a remarkable improvement in wearing comfort based on the improvement of textile materials and sensors. However, the washing of textiles may cause damage to the sensor and also has a risk of cross-infection. Respiratory monitoring based on radio frequency sensors still faces the problem of signal isolation interference. Compared to existing studies, this work broadens the focus of the current research, consisting of an anatomical and experimental analysis. We proposed a respiratory monitoring method based on pharyngeal impedance changes and designed a prototype respiratory monitoring system with head-mounted electrical impedance sensing for validation. The electrode location of this prototype was located near the mastoid bone of the head, which is located behind the human ear, and this location overlaps with a large number of wearable device wearing locations. After further optimization, it could be used as a promising solution for integrating helmets, glasses, and other devices, enabling sensorless respiratory monitoring, reusability, and eliminating cross-contamination. At the same time, the solution will not affect the permeability of breathing. Naturally, there are still many improvements to be made to our solution.

**Reliability:** We demonstrated that the system designed in this paper could accurately measure the respiration rate of 16 healthy young subjects in a laboratory environment. However, some differences exist between the laboratory environment and daily life scenarios, such as walking or cycling. Motion artifacts may influence the measurement results of respiration. Furthermore, we are not aware of the suitability of the system for patients with respiratory tract infections. There is a lack of evaluation of the effect of different age groups, BMI, gender, and other factors on the system. Therefore, we need further validation experiments to evaluate and improve the system.

**Miniaturization:** We designed a prototype of the miniaturized electrical impedance respiratory monitoring system. The prototype confirmed the feasibility of electrical impedance respiratory monitoring based on pharyngeal changes. The product size was 68 × 43 × 18 mm. Although it could be integrated into slightly larger helmets and other headed wearable devices, it would not be easy to integrate into small devices such as glasses. Currently, we use the impedance measurement scheme combining STM32F103RCT6 and AD5933, which can be further optimized in terms of power consumption. For example, we could use the lower-power msp430 series and an integrated circuit design, optimize the excitation frequency, and downsample to reduce power consumption and product size. Meanwhile, we used Ag/AgCl electrodes, which still have a considerable size and are not easy-to-use consumables. We can improve future research by exploring a dry electrode measurement.

## 6. Conclusions

In this paper, according to the effects of gas volume and conductance path changes in the respiratory tract on electrical impedance measurements in the head during respiration, we proposed a respiratory monitoring method based on electrical impedance sensing of pharyngeal changes. Firstly, we quantified and analyzed the electrical impedance changes generated by physiological changes in the pharynx during respiratory activity via a finite element simulation model. After that, we proposed a head-mounted respiratory monitoring scheme based on electrical impedance sensing and designed an electrical impedance respiratory sensing system that could be integrated into wearable devices such as helmets and glasses. Finally, we conducted controlled experiments to verify the effectiveness of the pharyngeal electrical impedance sensing system for extracting respiratory rate. We collected pharyngeal impedance data from all subjects in an inactive state in the laboratory and performed adaptive filtering for noise reduction. Compared with commercial respiratory monitoring devices, the correlation between the extracted respiratory waveforms of our proposed head-mounted electrical impedance respiratory monitoring system and those collected by commercial respiratory monitoring devices was as high as 0.9965. In addition, we analyzed the impedance data collected during respiratory monitoring among 16 subjects. Taking the commercial respiratory monitoring device as a reference, we found no significant difference in respiration rates measured from the head-mounted electrical impedance respiratory monitoring system by a paired *t*-test (*p* > 0.05). The results showed that the respiratory rates of all subjects were within the 95% confidence interval, indicating that our proposed head-mounted electrical impedance respiratory monitoring protocol had a good consistency with the commercial respiratory monitoring scheme. Therefore, the head-mounted electrical impedance respiratory monitoring method proposed in this paper is a promising solution for real-time continuous respiratory monitoring at work, for entertainment, and in daily life.

## Figures and Tables

**Figure 1 biosensors-12-00934-f001:**
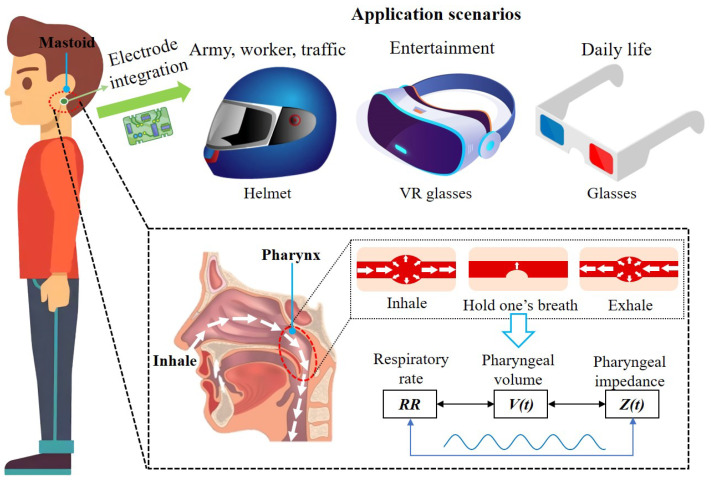
Head-mounted respiratory monitoring scheme.

**Figure 2 biosensors-12-00934-f002:**
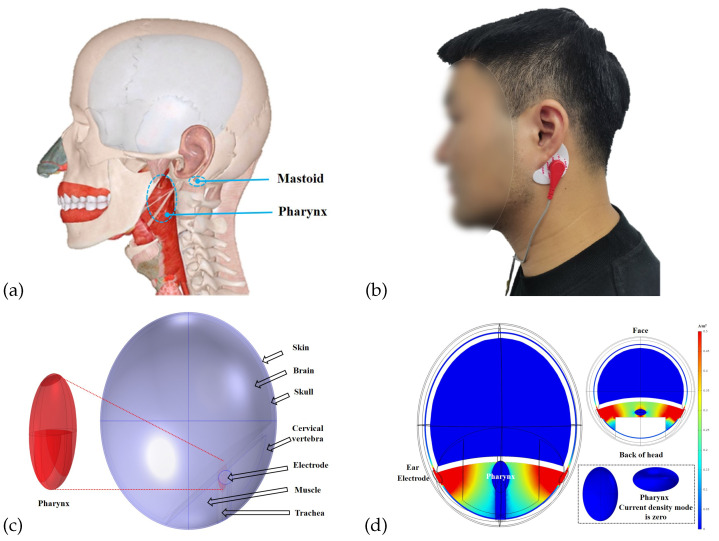
Construction of finite element model. (**a**) Anatomical structure of the head. (**b**) Schematic diagram of electrode adhesion in subjects. (**c**) Equivalent breathing simulation model of the head. (**d**) Head cross-sectional current density mode distribution.

**Figure 3 biosensors-12-00934-f003:**
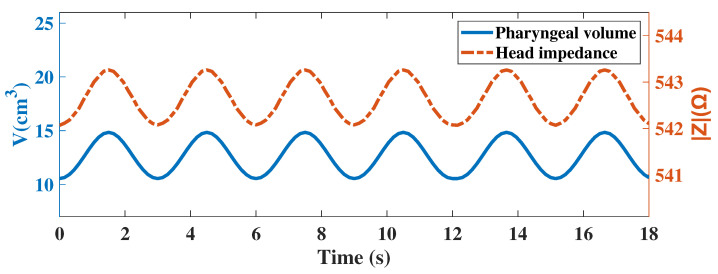
Relationship between pharyngeal volume and head impedance during respiration.

**Figure 4 biosensors-12-00934-f004:**
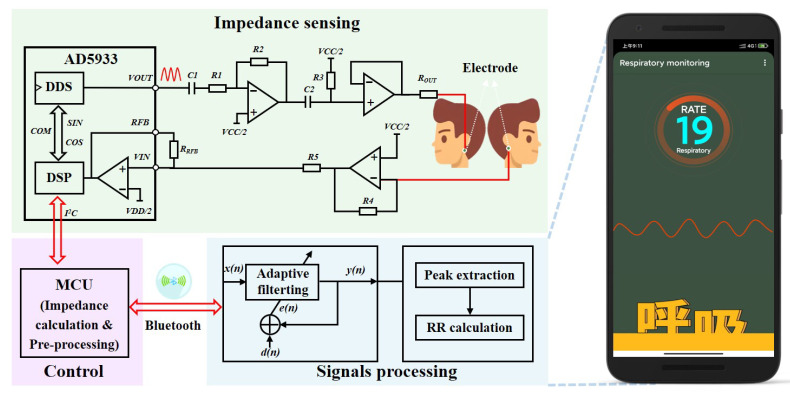
Schematic diagram of the electrical impedance sensing system for respiratory monitoring.

**Figure 5 biosensors-12-00934-f005:**
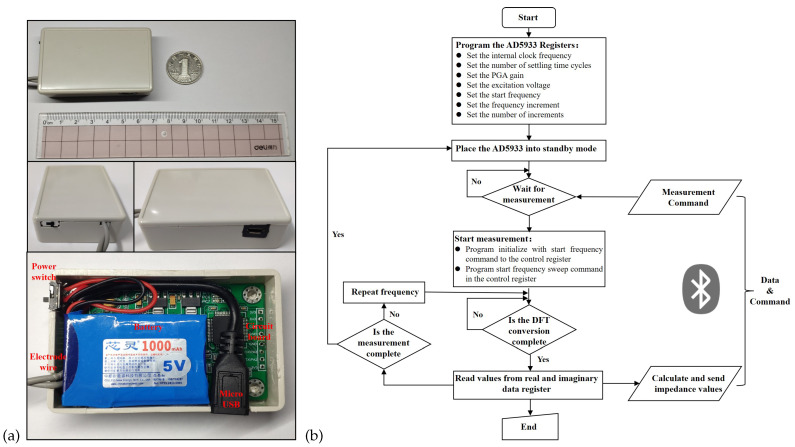
Display of electronic devices and software flow chart: (**a**) The physical diagram of the electronic device; (**b**) The flow chart of the embedded software.

**Figure 6 biosensors-12-00934-f006:**
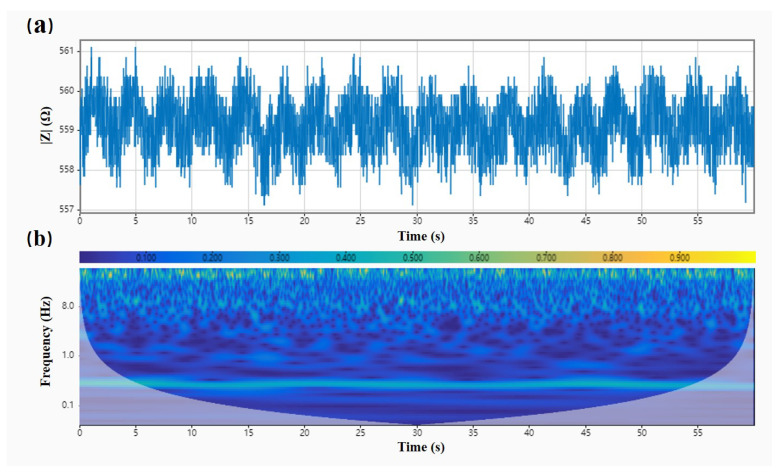
Impedance data of the head and their time–frequency distribution. (**a**) Subject one’s head impedance amplitude changes during breathing. (**b**) Head impedance time–frequency diagram during breathing in subject one.

**Figure 7 biosensors-12-00934-f007:**
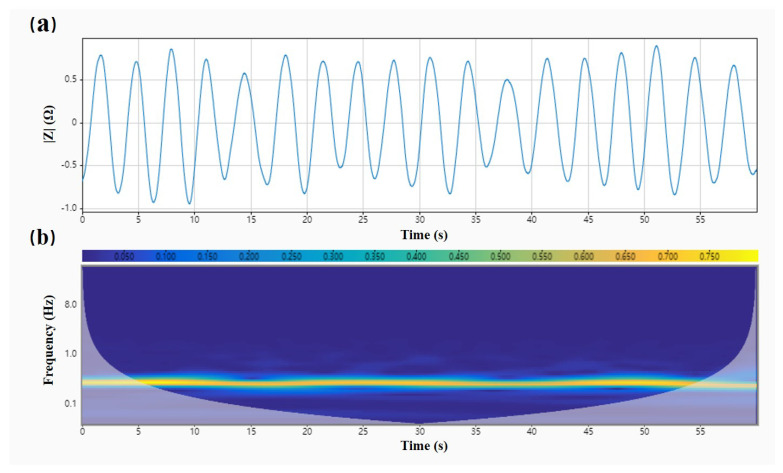
Respiratory waveform after adaptive filtering and its time–frequency plot. (**a**) Adaptive filtering and noise reduction to extract respiratory waveform based on subject one’s head impedance data. (**b**) Time–frequency plot of the extracted respiratory waveform based on subject one’s head impedance data with adaptive filtering and noise reduction.

**Figure 8 biosensors-12-00934-f008:**
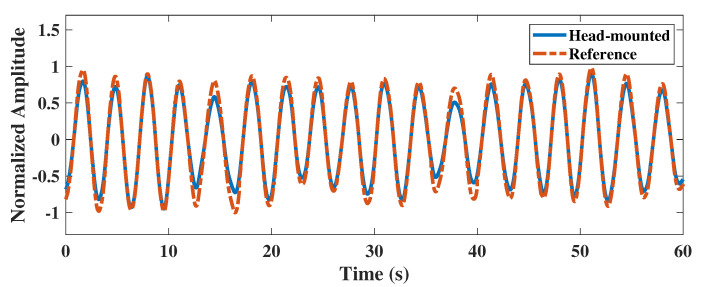
Comparison of respiratory waveforms measured by the two devices.

**Figure 9 biosensors-12-00934-f009:**
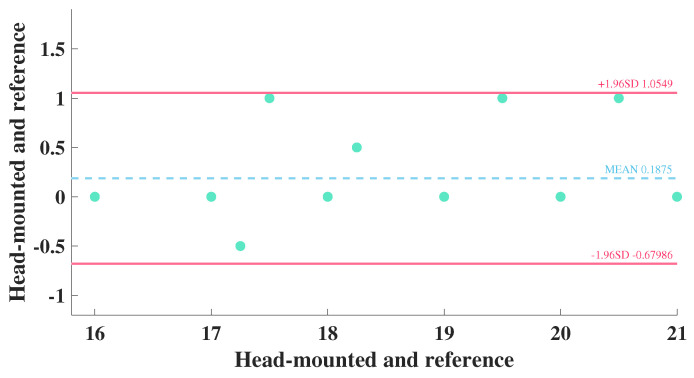
Bland–Altman distribution of respiration rates measured by the two device.

**Table 1 biosensors-12-00934-t001:** Parameters of simulation model.

Tissue	Tissue Thickness	Conductivity (S/m)	Relative Permittivity
Skin	thickness = 2 mm	4.33 × 10^−2^	1.63 × 10^2^
Skull	thickness = 10 mm	2.06 × 10^−2^	2.64 × 10^2^
Cervical vertebra	-	2.06 × 10^−2^	2.64 × 10^2^
Brain	-	1.28 × 10^−1^	5.46 × 10^3^
Muscle	-	3.52 × 10^−1^	1.01 × 10^4^
Pharynx (gas)	volume = 10.54∼ 14.84 cm${}^{3}$	0.00 × 10^0^	1.00 × 10^0^
Trachea (gas)	radius = 8 mm	0.00 × 10^0^	1.00 × 10^0^
Electrode	radius = 10 mm	5.00 × 10^5^	1.00 × 10^0^

**Table 2 biosensors-12-00934-t002:** Statistical information of the measurement results from the two devices.

Category	Values
Effective sample size	16
Mean (head-mounted)	18.781
Mean (reference)	18.594
Mean of the difference	0.188
Standard deviation	0.443
95% CI (Mean of the difference)	−0.048∼0.423
95% CI (Difference)	−0.680∼1.055
*t*	1.695
*p*	0.111
Coefficient of repeatability	0.917

**Table 3 biosensors-12-00934-t003:** Comparison of respiratory monitoring wearable devices.

Reference, Year	Type of Sensor	Wearing Style	Data Transmission and Processing	Reported Characteristics
[32], 2022	Flexible pressure sensor	Face mask	Wifi, PC, Matlab	Ultrathin self-powered, sensors affect the air permeability of the face mask.
[18], 2022	Textile capacitive sensor	Face mask	UART, mobile app	Disposable and consumable, lightweight handheld device.
[33], 2022	Pressure sensor	Waist belt	Wifi, SD card, PC	ASiT does not require calibration and is sufficiently sensitive, low air permeability.
[19], 2020	All-textile sensor array	Chest, abdomen, or wrist	Wirelessly, mobile app	Washable, high stability and comfort, textiles easily stained.
[34], 2019	Ultrasonic sensor	Abdomen-apposed rib cage	Bluetooth, mobile app	Wireless wearable measurement, relatively complex circuit, signal susceptible to interference.
[35], 2019	Radio frequency identification	Shoulder	Radiofrequency, PC	Passive radiofrequency identification, low power consumption, signal susceptible to interference.
[26], 2022	Bioimpedance	Chest patch	Bluetooth, LoRa, on device in real time	Real-world scenarios evaluation, chest movement interferes with the measurement signal.
This work	Bioimpedance	Head-mounted (mastoid)	Bluetooth, on device, and mobile app	Pharyngeal respiratory monitoring, comfortable to wear, easy to integrated and reusable.

## Data Availability

The data presented in this study are available on request from the corresponding author. The data are not publicly available due to privacy.

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
