# Peer review of "Feasibility Analysis and Implementation of Head-Mounted Electrical Impedance Respiratory Monitoring"

_biosensors, 2022, doi:10.3390/bios12110934_

Round 1
Reviewer 1 Report
The paper treats an actual subject.
The objective of the paper is easy to understand.
The paper is presented in a comprehensive manner.
The paper contains 28 applicable references.
BUT:
1. The "state of the art" (similar work) section is very poor presented - Are there on the market similar researches?
You must present a comparative analysis between your product and some others products...
2. You said that:
"We found that the respiration rates measured by the head-mounted electrical impedance respiratory monitoring system WERE NOT SIGNIFICANTLY DIFFERENT from those of commercial respiratory monitoring devices by the paired t-test (P>0.05)."
In this case, what are the advantages of your product? And what are the other products which you compare to?
3. Related to your design and implementation - you present only a schematic diagram...nothing more... You can present the REAL PRODUCT - images, some software code of your application...
4. You made a case study on 16 persons... I appreciated this study. In the future you can add more persons to your study.
5. What makes your product different? I repeat, I see no comparative analysis, I see no other products on the market... I see a poor validation of your product (16 persons)...
Your work is interesting but it is not sufficiently explained to be accepted to BIOSENSORS.
You can resubmit to other journal.
6. What are the limitations of your research?
No limitation is specified...
Author Response
Response to Reviewer 1 Comments
Point 1: The "state of the art" (similar work) section is very poor presented - Are there on the market similar researches?
Response 1: We are very thankful for the reviewers' suggestions, and we did not do enough preliminary comparison work. For the latest wearable respiratory monitoring devices, we have added a comparison of commercially available medical devices and textile-based respiratory monitoring studies in the introduction section (lines 36-47) and added Table 3 for comparison.
Point 2: You said that:
"We found that the respiration rates measured by the head-mounted electrical impedance respiratory monitoring system WERE NOT SIGNIFICANTLY DIFFERENT from those of commercial respiratory monitoring devices by the paired t-test (P>0.05)."
In this case, what are the advantages of your product? And what are the other products which you compare to?
Response 2: The reviewers' suggestions are greatly appreciated. To verify the feasibility of the pharyngeal impedance-based respiratory frequency extraction scheme proposed in this paper, we used a paired t-test of Bland-Altman distribution plots for consistency assessment. A control study with a commercial respiratory monitoring device (SNORE CIRCLE, Y20) showed that the electrical impedance sensing-based respiratory monitoring system designed in this paper is capable of accurately monitoring the respiratory activity of subjects. For more details about the advantages of our product compared with other products, see lines 324-344.
Point 3: Related to your design and implementation - you present only a schematic diagram...nothing more... You can present the REAL PRODUCT - images, some software code of your application...
Response 3: We appreciate the reviewers' guidance. Since this study is currently patent pending, more details were not shown in the early stage. Indeed, as a paper publication, we need to show more information. Therefore, we have added the real product diagram and the software flowchart display in section 3.2.3 of the main text, as shown in Figure 5.
Point 4: You made a case study on 16 persons... I appreciated this study. In the future you can add more persons to your study.
Response 4: We thank the reviewers for their affirmation. Currently, this paper focuses on the feasibility study of a respiration rate extraction method based on pharyngeal impedance changes. The focus of the paper is to propose an equivalent model for the head (pharynx) impedance changes during respiration, and to verify the effectiveness of the method through simulation and experiments. Therefore, a case study was conducted in a laboratory setting for 16 subjects only in the preliminary stage. We will carry out more studies after further optimization and improvement of the system, such as the reliability of the system, which is a concern of the reviewers.(lines 345-363)
Point 5: What makes your product different? I repeat, I see no comparative analysis, I see no other products on the market... I see a poor validation of your product (16 persons)...
Response 5: In response to the reviewers' comments, we have added a discussion section with a comparative discussion of existing products and our products.
Existing wearable respiratory monitoring devices have made significant progress in wearing comfortably and coping with complex monitoring scenarios while ensuring high-accuracy respiratory rate measurements. Table III summarizes and compares some of the latest wearable respiratory monitoring systems. The current research points focus on the improvement of new materials and sensors. Respiratory monitoring sites include the mouth, nose, chest, and abdomen, with little exploration of respiratory monitoring in the pharynx. There is a remarkable improvement in wearing comfort based on the improvement of textile materials and sensors. However, the washing of textiles may cause damage to the sensor and also the risk of cross-infection. Respiratory monitoring based on radio frequency sensors still faces the problem of signal isolation interference. Compared to existing studies, this work broadens the focus of the current research, consisting of anatomical and experimental analysis. We have proposed a respiratory monitoring method based on pharyngeal impedance changes and designed a prototype respiratory monitoring system with head-mounted electrical impedance sensing for validation. The electrode location of this prototype is located near the mastoid bone of the head, which is located behind the human ear, and this location overlaps with a large number of wearable device wearing locations.After further optimization, it can be used as a promising solution for integrating helmets, glasses, and other devices, enabling sensorless respiratory monitoring, reusability, and eliminating cross-contamination. Naturally, there are still many improvements to be made to our solutions.
Point 6: What are the limitations of your research? No limitation is specified...
Response 6: At present, we have only presented the design and verification of the prototype product. We must acknowledge that there are still many shortcomings in this study. Therefore, in the discussion in Section 5, a description of limitations is added and future work points out areas for improvement in the future.
Reliability: We have demonstrated that the system designed in this paper can accurately measure the respiration rate of healthy young subjects in a laboratory environment. However, some differences exist between the laboratory environment and daily life scenarios, such as walking or cycling. Motion artifacts may influence the measurement results of respiration. Furthermore, we are not aware of the suitability of the system for patients with respiratory tract infections. There is a lack of evaluation of the effect of different age groups, BMI, gender, and other factors on the system. Therefore, we need further validation experiments to evaluate and improve the system.
Miniaturization: We have designed a prototype of the miniaturized electrical impedance respiratory monitoring system. The prototype has verified the feasibility of electrical impedance respiratory monitoring based on pharyngeal changes. The product size is 68*43*18mm. Although it can be integrated into slightly larger helmets and other headed wearable devices, it is not easy to integrate into small devices such as glasses. Currently, we use the impedance measurement scheme combining STM32F103RCT6 and AD5933, which can be further optimized in terms of power consumption. For example, we can use lower power msp430 series and integrated circuit design, optimize the excitation frequency, and downsampling to reduce power consumption and product size. Meanwhile, we use the Ag/AgCl electrode, which still has a shortage of considerable size and easy consumables. We can improve future research by exploring the dry electrode measurement.

Reviewer 2 Report
The manuscript present and interesting development for a noninvasive respiratory monitoring system.
Some comments for this review are:
• The authors state in line 117: "An excitation current with an amplitude of 1 mA and a frequency of 50 Hz was added to the electrode below the mastoid bone to simulate six cycles of respiratory activity according to equation 1."
How does an excitation of 50 Hz simulate six cycles of respiratory activity? Please rephrase.
• It is not clear from the described protocol whether the authors used dry or gel AgCl electrodes. The information is crucial for understanding the impedance measurements.
• The authors use an excitation of 50 kHz. Why this specific frequency? Have the authors studied other frequencies and concluded this was optimal?
• It is important to state that the AD5933 and the front-end analog circuitry (as presented in Fig.5) will render results *proportional* to the admittance of the subject-under-test. This means that for the actual value of the impedance, calibration and inversion of the values need to be done. I understand that it might not be necessary for extracting the respiratory frequency, but I think the authors should address this issue.
• The authors state in Conclusions that they studied the effects of gas volume and conductance paths on the electrical impedance being measured. Actually that was not done in this paper; (as stated before in the manuscript, the authors base their study in a previous work of Tavanto et al. [25].) The authors did developed a system harnessing the effect, and validated the hypothesis using FEM and a clinical trial.
• A few points the authors should mention as future research:
1. Effects of respiratory tract infections in the results.
2. How often the electrodes should be replaced to avoid changes in the measured values.
3. How the BMI, age, gender, and other parameters may or not influence the results (including the selection of the working frequency).
• Are the authors going to publish their system as an Open Source project or are they filing for a patent (in which case it should be stated in the Conflict of Interest statement)?
Author Response
Response to Reviewer 2 Comments
Reviewer 2: The manuscript present and interesting development for a noninvasive respiratory monitoring system.
Response: We are very grateful to the reviewers for your recognition and suggestions. It is true that the details in many parts of our paper are not described very well. With the sincere suggestions of the reviewers, we have revised it as follows:
Point 1: The authors state in line 117: "An excitation current with an amplitude of 1 mA and a frequency of 50 Hz was added to the electrode below the mastoid bone to simulate six cycles of respiratory activity according to equation 1."
How does an excitation of 50 Hz simulate six cycles of respiratory activity? Please rephrase.
Response 1: Thank you very much to the reviewers for their carefulness. We have reworded the description to address the issue.
We modified the volume of the pharynx according to equation 1, which was used to simulate six cycles of respiratory activity. Simultaneously, an excitation current of 1 mA and 50 kHz was added to the electrode below the mastoid bone. (lines 126-129)
Point 2: It is not clear from the described protocol whether the authors used dry or gel AgCl electrodes. The information is crucial for understanding the impedance measurements.
Response 2: The reviewers' guidance is much appreciated. We used a crescent-shaped gel silver chloride electrode (Jing dian yi kang, LD-1) and added Figure 2(b) with additional descriptions in the text.
After the subject was seated at the assigned position, the skin below the mastoid bone on both sides of the subject's head was wiped with alcohol swabs, and Crescent-shaped gel Ag/AgCl physiotherapy electrodes (Jing dian yi kang,LD-1) were attached to the area, which was shown in Fig. 2(b).(lines 259-262)
Point 3: The authors use an excitation of 50 kHz. Why this specific frequency? Have the authors studied other frequencies and concluded this was optimal?
Response 3: We chose an excitation of 50 kHz, referring to Schwartz S's findings that muscles tend to be most responsive around 50 kHz. Of course, for the measurement of head impedance, it is a good direction to further investigate the applicability of other frequencies, which we will further investigate in the future.
Schwartz S, Geisbush TR, Mijailovic A, Pasternak A, Darras BT, Rutkove SB. Optimizing electrical impedance myography measurements by using a multifrequency ratio: a study in Duchenne muscular dystrophy. Clin Neurophysiol. 2015 Jan;126(1):202-8. doi: 10.1016/j.clinph.2014.05.007.
Point 4: It is important to state that the AD5933 and the front-end analog circuitry (as presented in Fig.5) will render results *proportional* to the admittance of the subject-under-test. This means that for the actual value of the impedance, calibration and inversion of the values need to be done. I understand that it might not be necessary for extracting the respiratory frequency, but I think the authors should address this issue.
Response 4: The reviewers are very knowledgeable about the AD5933 and have a very rigorous academic attitude. In response to the reviewers' comments, we have added the design derivation and calibration of the AD5933 and front-end analog circuit in the article, which is detailed in lines 181-196 in the article
Point 5: The authors state in Conclusions that they studied the effects of gas volume and conductance paths on the electrical impedance being measured. Actually that was not done in this paper; (as stated before in the manuscript, the authors base their study in a previous work of Tavanto et al. [25].) The authors did developed a system harnessing the effect, and validated the hypothesis using FEM and a clinical trial.
Response 5: Thank you very much for the reviewers' guidance. We have reworked the description in the conclusion section, as detailed in lines 365-368.
Point 6: A few points the authors should mention as future research:
1.Effects of respiratory tract infections in the results.
2.How often the electrodes should be replaced to avoid changes in the measured values.
3.How the BMI, age, gender, and other parameters may or not influence the results (including the selection of the working frequency).
Response 6: We greatly appreciate the reviewers' suggestions. We have added a discussion section in the paper to compare the existing products and to explain the limitations of our study and future research ideas. Such as Table 3, lines 324-363
Point 7: Are the authors going to publish their system as an Open Source project or are they filing for a patent (in which case it should be stated in the Conflict of Interest statement)?
Response 7: We are very grateful for the reviewers' suggestions. Since this study is currently patent pending, more details were not shown in the early stage. Indeed, as a paper publication, we need to show more information. Therefore, we have added the real product diagram and the software flowchart display in section 3.3 of the main text, as shown in Figure 5.

Reviewer 3 Report
The authors developed a head-mounted electrical impedance respiration monitoring system, which based on electrical impedance sensing and integrated with adaptive filtering and respiratory tracking algorithms in to an APP for the mobile phone. Since the respiration leads to changes in gas volume and associated muscle morphology in the body, the electrical impedance of muscles can be measured as the the crucial indicators for monitoring human physiological health. The theoretical analysis is well done and the comparisons between theoretical and experimental results is good to prove the effectiveness. This paper is well written, and it could provide valuable concept to researchers who are interested in this topic. This reviewer recommends the manuscript is good to be accepted
However, some part is better to described more clear, for example, how heave of the monitoring system, especially for longer time wearing. Any uncomfortable will be caused. Also how long time the system can be used, what the power supplier used. What are the barratry and wireless system, etc.
Author Response
Response to Reviewer 3 Comments
Reviewer 3:The authors developed a head-mounted electrical impedance respiration monitoring system, which based on electrical impedance sensing and integrated with adaptive filtering and respiratory tracking algorithms in to an APP for the mobile phone. Since the respiration leads to changes in gas volume and associated muscle morphology in the body, the electrical impedance of muscles can be measured as the the crucial indicators for monitoring human physiological health. The theoretical analysis is well done and the comparisons between theoretical and experimental results is good to prove the effectiveness. This paper is well written, and it could provide valuable concept to researchers who are interested in this topic. This reviewer recommends the manuscript is good to be accepted.
Point: However, some part is better to described more clear, for example, how heave of the monitoring system, especially for longer time wearing. Any uncomfortable will be caused. Also how long time the system can be used, what the power supplier used. What are the barratry and wireless system, etc.
Response: We are very grateful to the reviewers for your recognition and suggestions. It is true that the details in many parts of our paper are not described very well. With the sincere suggestions of the reviewers, we have revised it as follows:
- About the system's physical product and more design details, we have added Figure 5 to show and the derivation of Equations 3-5, see line 181-196for details.
- We used a crescent-shaped gel silver chloride electrode (Jing dian yi kang, LD-1), which can be firmly affixed under the subject's mastoid bone with minimal effect on the subject. Detailed modification 2(b) and lines 259-261 。
- We used STM32F103RCT6 from STMicroelectronics as the microprocessor unit and Bluetooth 4,0 as the communication method. The hardware PCB was designed and fabricated by EasyEDA (https://lceda.cn/) from Shenzhen Jalitron Technology Development Co. The battery is a 5V1A Boosting Lithium Batteries (1000mAh) from ZONCELL. After the initial test, it can be used for 10 hours. Future optimization in power consumption, we will use msp430 series for design, electrical impedance sensing module can be optimized by integrated circuit (IC) design to achieve more miniaturization and low power consumption to meet the wearable demand.Details of the modifications are reflected in lines 240-250of the paper.

Round 2
Reviewer 1 Report
The paper treats an actual subject.
The objective of the paper is easy to understand.
The paper is presented in a comprehensive manner.
The paper contains applicable references.
Proposals:
1. The "state of the art" (similar work) section can be improved -
You can present a more detailed comparative analysis between your product and some others products...
2. You can present better what are the advantages of your product...
3. The flow desgin, implementation, validation should be very clear!!!
16 peersons to validate your product... only! In the future you can add more persons to your study.
Your work is interesting and it can be accepted to BIOSENSORS.
Author Response
Response to Reviewer 1 Comments
Reviewer 1:
The paper treats an actual subject.
The objective of the paper is easy to understand.
The paper is presented in a comprehensive manner.
The paper contains applicable references.
Your work is interesting and it can be accepted to BIOSENSORS.
Response: We are very grateful to the reviewers for your recognition and suggestions. It is true that the details in many parts of our paper are not described very well. With the sincere suggestions of the reviewers, we have revised it as follows:
Point 1: The "state of the art" (similar work) section can be improved -
You can present a more detailed comparative analysis between your product and some others products...
Response 1: We appreciate the reviewers' suggestions. For the comparative product analysis, we have added a column "Reported Characteristics" in Table 2 to compare the characteristics of similar works.
Point 2: You can present better what are the advantages of your product...
Response 2:In order to better reflect the advantages of our product, we briefly introduce the features of our work in the introduction section(lines 70-78). The advantages of our work compared with similar studies are analyzed in the discussion section.(lines 327-342)
Point 3: The flow desgin, implementation, validation should be very clear!!!
16 peersons to validate your product... only! In the future you can add more persons to your study.
Response 3: The reviewers are very rigorous in their academic thinking. Regarding the flow, implementation, and validation of the paper. We have redescribed the structural arrangement of the paper in the introduction section to make it easier for readers to understand the work in this paper(lines 79- 86). The description has also been revised in several places in the text.The limitations about the number of people studied are also discussed and prospected in the discussion section.(lines344-345,349-351 ).
